# Searching for the Optimal Method of Financing Hospital Emergency Departments—Comparison of Polish and Selected European Solutions

**DOI:** 10.3390/ijerph19031507

**Published:** 2022-01-28

**Authors:** Anna Tyrańska-Fobke, Marlena Robakowska, Daniel Ślęzak, Katarzyna Pogorzelczyk, Andrzej Basiński

**Affiliations:** 1Department of Public Health & Social Medicine, Medical University of Gdańsk, 80-210 Gdańsk, Poland; mrobakowska@gumed.edu.pl; 2Department of Medical Rescue, Medical University of Gdańsk, 80-210 Gdańsk, Poland; andrzej.basinski@gumed.edu.pl; 3Independent Researcher, 8200 Aarhus, Denmark; kpogorzelczyk@gmail.com

**Keywords:** hospital emergency departments, financing, overcrowding

## Abstract

Hospital emergency departments are units of the State Medical Rescue system in Poland, which was established to help people in a state of a health emergency. The aim of this study is to develop an optimal method of financing emergency departments in Poland. The study used Polish data from 2016–2019 on the financing of services at the Clinical Emergency Department of the University Clinical Center in Gdańsk. For benchmarking and mathematical modeling, data for the Czech Republic, Germany and Latvia was used. The results of the analysis shows significant differences, to the disadvantage of Clinical Emergency Department, between the potential contract values in the tested models and the actual amounts of funds transferred by the National Health Fund Pomeranian Voivodeship Branch for the activities of Clinical Emergency Department under the concluded contracts. The introduction of co-payment on the part of patients reporting to the emergency departments with minor ailments that do not require hospitalization generates financial revenues, but does not significantly improve the financial results of the analyzed ward. However, it may be educational for patients in terms of raising their awareness of the correct place to seek assistance in the event of a sudden illness.

## 1. Introduction

Hospital emergency departments (EDs) are units of the medical rescue system, which was established to help people in a state of sudden health emergency. A situation of sudden threat to health and life may concern any person, regardless of gender, age, health condition or comorbidities.

In order to enable emergency departments to properly perform the tasks for which they have been established, it is important to have them properly and adequately financed and supported by state institutions. Staffing problems and difficult access to health care services at lower levels, i.e., in primary health care and outpatient specialist care, mean that patients who do not require immediate medical assistance due to health emergencies also seek assistance from emergency departments. Hence, it is highly important to look for solutions aimed at improving the functioning and financing of EDs, so patients in health and life-threatening situations could benefit from professional help at these departments as soon as possible [1].

A large number of different models for the funding the activities of hospital emergency departments exist worldwide. Some are presented below.

In Poland, the financing of hospital emergency departments is based on a daily flat rate calculated on the basis of a formula consisting of four elements. The first is the base rate based on the minimum equipment, organization and departmental minimum human resources requirements of $1180. The rate may be adjusted depending on the ward’s resources. Another component is the product of the daily rate and the indicator of fulfilled additional conditions (organizational and staffing), expressed as a percentage. The third component of the rate takes into account the number of patients classified in the appropriate category of the patient’s health, based on the performed medical procedures specified in the list of medical procedures according to ICD-9, and the weight assigned to each category. 

In the Czech Republic, emergency care is reimbursed by health insurance according to the $0.044 point-value list of medical procedures and is included in the limit set for the hospital as a whole. This limit is set as a fixed reimbursement ceiling. In addition, there is a $4.4 surcharge for each patient transported to the hospital emergency department by the ambulance service and a surcharge for nursing care for each patient. The total amount for the care of patients in emergency situations is limited by the maximum reimbursement, which is based on the verified number of services provided during the reference period and is dependent on the center’s reference level. In addition, hospitals also receive a flat-rate emergency department subsidy of $118,800 per year. In the Czech Republic emergency departments are financed in the same way as other hospital departments. Additionally, if the patient uses the ambulance services, he/she pays a regulatory fee of $4. It is similar in the case of reporting to the emergency department. This fee is canceled if the patient is hospitalized on the basis of a doctor’s decision [2].

Medical services in the Latvian EDs are financed in addition to the total remuneration for hospitals. The subsidies for hospitals for ED management consist of three elements. The first is a flat-rate fixed remuneration for preparedness (including remuneration of doctors and medical staff), as well as maintenance costs (medicines, overheads, administration costs, depreciation). The value of the flat rate fixed amount is fixed and does not depend on the hospital’s reference category. The second is a bonus for observing a patient for up to 24 h. The patient can be followed up in the observation bed for up to 24 h. A doctor assesses the patient’s condition and makes a decision on the strategy for his/her further treatment, as well as on hospitalization or possible continuation of treatment on an outpatient basis. The follow-up premium is set according to a variable tariff depending on the hospital’s reference category. The last one is remuneration according to a fixed tariff for services provided in first aid offices located in the ED, mainly for examining patients. In addition, there is patient co-payment for some diagnostic tests according to the statutory price list [3].

There are significant regional differences between the 16 German states (Länder) in terms of regulation, organization and financing of after-hours health services, emergency medical services and emergency care. Services provided in the German emergency departments are not financed separately from other general hospital services. Full and partial hospital services are remunerated on the basis of the diagnosis-related group system (adaptation of the Australian version of the “diagnosis-related group”, DRG). In the case of a flat rate, the remuneration for a specific disease entity and its treatment is calculated in a specific range of the length of the patient’s stay in the hospital. Basically, the lump sum price for the treatment is obtained by multiplying the relation of the valuation of the relevant group of diagnosis-related groups by the base value. The diagnosis-related group calculations are based on actual performance data for all hospitals and actual cost data from a sample of hospitals voluntarily submitting their cost data. In addition, hospitals receive public funds to maintain the minimum statutory level of employment of nurses and carers/nurses. Patients’ co-payments for the stay in the hospital itself (includes stay, meals and other costs related to staying in the hospital) also contribute to the funding of emergency departments [4].

The study attempts to find an answer to the question of how to modify the principles of ED financing in Poland, using the experience of other countries, so that they correspond to the actual financial needs of these departments, taking into account their operating conditions. The work focuses on the analysis of both financial and non-financial solutions from various countries of the world and the possibility of their adaptation to Polish conditions.

## 2. Materials and Methods

### 2.1. Materials

The first part of Polish data comes from the CGM CLININET HIS IT system of the clinical hospital in Gdańsk and concerns the characteristics of patients and procedures performed in the emergency department of this hospital in 2017–2019. CGM CLININET is a comprehensive information system supporting the processes of treatment and management in a medical facility. It enables the management of treatment processes and medical information. The system consists of many specialized functional subsystems and modules, including the HIS module.

From the client system for analyzes, data on the number of patients admitted to CED in individual years in the analyzed period was obtained, along with information on their health status broken down into individual categories of patient’s health status in ED (from I to VI), which constitute the basis for reporting and calculating the benefits provided by CED.

Data for the valuation of health services provided was obtained from publicly available databases.

Detailed data on the methods of financing EDs in countries with which cross-border cooperation in patient care takes place were also important for the analysis. Data from the Czech Republic, Germany and Latvia were included, either due to the similarity of regions, population, way of life, or economic data in comparison to Poland. Detailed data on the financing of EDs in these countries were obtained by means of correspondence with relevant government units and foreign payers.

### 2.2. Methods

Then, on the basis of data from the analyzed countries, four ED financing models were constructed. Later, the previously prepared data on the number of patients in individual categories of patient health status in CED was used to simulate ED funding in the developed models. These models include:

1.Model based on the German system (model A):

Funds for ED = payment for procedures in the reporting period according to the rates applicable in the reporting period (including salaries) + patient’s co-payment for the stay (per-night rate)—applies to all patients, regardless of the length of stay;

2.Model based on the Czech model (model B):

Funds for ED = fee for procedures in the reporting period according to the rates applicable in the reporting period (including salaries) + lump sum for each patient transported to the ED by the emergency medical service ($4.42) + patient’s co-payment if he/she presents himself/herself and is not hospitalized ($3.98);

3.Model based on the Latvian model (model C):

Funds for ED = current daily flat rate + fee for patients undergoing observation (fee per person-day, but unlike the model based on the German model, it is paid from public funds);

4.An additional model based on the compilation of the above models (model D):

Funds for ED = the current daily lump sum + co-payment from each patient who is transported by the emergency medical service or who reports to the ED, but their condition does not require hospitalization.

The common denominator is the presence of legally sanctioned co-payment on the part of the patient, which at the same time is a factor that distinguishes the models from the current Polish healthcare system. Therefore, it became justified to construct and verify models including a co-payment on the part of the patient. Similarly, the creation of a model that uses payment for medical procedures performed, and not a lump sum, was justified in the collected data on the methods of financing ED in the analyzed countries. In this work, the estimates of the value of medical procedures provided by a CED were made in individual categories of the patient’s health in the ED based on the valuations used for outpatient specialist care and inpatient treatment. Comparisons of the estimated value of services provided in the CED were made according to the average values of the individual categories of the patient’s health in the ED and the value of contracts based on daily lump sums granted to the CED by the NHF in the reporting periods in 2017–2019. 

In addition, for the correct performance of the calculations, it was also necessary to determine the values for following variables:Value of the person-day stay in the ED: Due to the fact that it is not possible to estimate the value of a person-day stay in the CED on the basis of the available regulations and data, the calculation was based on the valuation of the part determined by the unit itself in the form of price lists valid for a given year, i.e., an average of $18.20 per hour of stay. In the case of model A verification, the payment for the stay concerned each patient in the ED. However, in the case of model C, the payment concerned only patients requiring observation in the ED UCC, i.e., all patients qualified in a given period to category III (extended diagnostic imaging, monitoring of basic vital functions, intravenous or intraosseous pharmacotherapy, small outpatient surgery, invasive examination), category IV (activities related to the maintenance of vital functions, extended diagnostics, intravenous infusions, endoscopy, resuscitation), category V (one-day hospitalization of a patient in ED—monitoring vital functions, extended diagnostic imaging) and category VI (one-day hospitalization of a patient at the Intensive Therapy).Lump sum for each patient transported to UCC ED by the emergency medical services: The lump sum was adopted at the level analogous to the one in force in the Czech Republic, which was converted into Polish zlotys at the average annual exchange rate of the National Bank of Poland [5] for the individual analyzed periods and amounts to $43.18 on average.The rate of co-payment for each patient who came to UCC ED independently and without a referral and did not require hospitalization: The rate was adopted at a level analogous to that applicable in the Czech Republic, which was converted into Polish zlotys at the average annual exchange rate of the National Bank of Poland [5] for the individual analyzed periods and amounts to $3.9 on average.The rate of co-payment for each patient who was transported to CED by the emergency medical services or who came to UCC ED independently and without referral and did not require hospitalization. The rate was adopted at the level analogous to that described above, i.e., $3.9.

Based on all the data presented in this chapter, the proposed models for financing services in ED were verified and split into the number of medical procedures provided by UCC ED and provided patients in the period 2017–2019.

## 3. Results

The analysis was based on the estimated values of services provided in the reporting periods in 2017–2019 in the ED of the Clinical Hospital of Gdańsk (UCC), calculated according to the average value of individual health categories of the patient in the ED, determined on the basis of the unit valuation used for outpatient specialist care and inpatient treatment. The data is presented in Table 1. 

The summary values obtained in Table 1 show a significant increase in the analyzed period, especially throughout 2019. The sources of these significant differences can be found in the increase in the number of patients in individual categories of patient’s health in 2017–2019. The total values obtained in Table 1 were compared with the actual value of the daily lump sums for the UCC in accordance with the agreements with the National Health Fund (NHF) Pomeranian Voivodeship Branch (PVB). The result of the analysis is presented in Table 2.

The value of the NHF contracts with the UCC in 2017–2019 on the basis of daily lump sums varied, but showed an upward trend in subsequent periods covered by the contract. Based on the data contained in Table 2, it can be concluded that, despite the systematic increase in the value of daily lump sums for the UCC in the analyzed period, the values of contracts with the NHF were much lower than the estimated value of services provided in the UCC, according to the average value of patient severity category in the ED in the reporting periods determined on the basis of the above-described valuation used for outpatient specialist care and inpatient treatment. Significant differences, reaching as much as 72%, can be observed especially in 2019. The average difference in the analyzed period was 54%. The summary data shows that between 2016 and 2019, the total number of patients in the UCC increased by 31%. On the other hand, the value of contracts with the NHF for health services provided in Clinical Hospital in Gdańsk ED in 2016–2019 increased by 35%. Based on the data and calculations presented above, calculations were made for the models proposed in the paper.

Based on the data contained in Table 3, it can be concluded that all the proposed models of financing services in the ED result in higher values than the value of contracts with NHF for the analyzed years 2017–2019. Detailed differences between the values obtained in individual models in relation to the value of contracts determined on the basis of a daily flat rate are presented in Table 4.

The data presented in Table 4 illustrate in more detail the order of magnitude of the differences between the values of contracts for the provision of services in the UCC concluded with the NHF in 2017–2019 and the estimated values of hypothetical contracts concluded based on the assumptions of the analyzed models. For 2017 and 2018, the differences between model B and the actual value of the UCC contracts with the NHF remain under 100%. On the other hand, in 2019, the largest difference is observed in the A, B and C models. In the entire analyzed period, the greatest differences between the models and the NHF contract occur with model A, which generates values on average more than 600% higher. Model D, on the other hand, maintains a constant, slight difference of 3% in the entire analyzed period.

## 4. Discussion

Therefore, it can be concluded that the change of the current method of financing the ED from a daily lump sum to the settlement based on individual services provided in the ED, similar to the German solutions, will result in a favorable change in the financial result of the examined CED, and thus an improvement in the financial result of the entire healthcare entity. It is also worth bearing in mind that this was still present in hospital treatment by 2017, when the point evaluation of individual services or groups of services gave way to the lump sum system for basic hospital healthcare services under the so-called hospital networks. Therefore, the introduction of a solution replacing the daily lump sums in the emergency department with the valuation of individual services provided would be beneficial, but it may prove impossible to implement due to the formula of lump sum financing dominating hospital treatment funding.

In the light of the above, the only applicable model among those verified in this study is the proposed additional model assuming the current daily lump sum for the ED increased by the co-payment from patients reporting to the ED and not requiring hospitalization at the rate applied in the Czech Republic.

However, as the analysis shows, this solution increases the total value of the contracts in the analyzed emergency department only by 3%, which does not compensate for the related financial losses. The use of a solution, similar to the Czech Republic model, where a patient co-pays for health services provided in the ED, which does not result in hospitalization, will not significantly improve the financial result of the examined ED, thus improving the financial condition of the entire healthcare entity.

First of all, it should be noted that in the foreign healthcare systems discussed in this paper, the common denominator is the presence of legally sanctioned patient co-payment, which at the same time distinguishes them from the Polish health care system.

However, it is worth bearing in mind that in the literature on the subject, patient co-payment for health services is perceived as the so-called a “barrier mechanism” restricting access to medical services. However, its effectiveness in reducing the total number of procedures performed, and thus also the potential reduction of the burden on the entire health care system, has not been unequivocally and definitively confirmed so far [7].

Currently, a potential reduction in the number of hospitalizations in ED, and thus an improvement in their financial situation, is seen, inter alia, in improving access to medical care within primary health care. Policy makers in many countries aim to improve the functioning of primary care in order to reduce the burden on emergency health services [8]. This is one of the main strategic goals of the health policy of many countries, as the limited financial resources of the health care system require rational management [9].

In the United Kingdom it has been shown that, in 2012–2013, as many as 26.5% of requests for assistance in the units of the emergency medical system were preceded by unsuccessful attempts to obtain medical help under primary health care. Their cost was £5.77 million [10]. The data analyzed in another study covering the British hospital treatment sector in 2011–2012 indicate that implementing improvements in access to primary care doctors resulted in a decrease in the proportion of medically unjustified ED admissions. On the basis of the obtained results, Cowling et al. concluded that better cooperation between the primary care physician and the physician in charge of the patient in the hospital might reduce the total costs of inpatient treatment while improving the quality of health care and treatment outcomes [10]. Other scientists in studies from the US and Canada also came to similar conclusions [11].

Moreover, the study conducted by Gill et al. showed that statistically significantly more often patients who work well with their primary care physicians on a daily basis asked them for advice before going for ED help, especially when they themselves had doubts as to whether they needed urgent assistance [12].

Another direction of activities aimed at improving the financial condition of ED are various attempts to provide a special form of outpatient care to high-risk and high-cost patients (HRHC patients) who are admitted to ED more often than other patients. So far, several pilot projects have been described and the results have proved very promising in terms of budgetary implications for hospital ED [13,14,15,16]. There is also evidence supporting the benefits of involving other people, not only medical professionals, in outpatient support of seriously ill patients. The team, led by Enard, carried out a pilot study that resulted in reductions in the number of ED interventions and thus lower costs of care. The obtained savings turned out to be higher than the costs of introducing the discussed solution [17].

Another way to reduce the financial problems of EDs with unconfirmed effectiveness is the introduction of telephone advice. Some researchers show a reduction in the number of ED visits thanks to such advice, and some conclude that the use of such a solution only postpones the patient’s reporting to the ED [18].

As a summary, it is worth citing Raven et al., who aimed to develop a systematic review of publications analyzing the impact of various pilot programs dedicated to improving the functioning of emergency departments on their actual costs before and after the interventions. However, the researchers concluded that due to the limited amount of available data, it is not possible to reach unambiguous conclusions [19]. In addition, it is also worth paying attention to the fact that many available publications assess the problem of the excessive load on both the ED and other units of the emergency medical system in total. Hence, problems arise in the interpretation of the results of such studies [20]. The common denominator of such studies is limited to the conclusion that health services provided in ED are the main cost-intensive element in the emergency medical system [21].

Another broad area of research and analysis involves improvements in the management, organization and functioning of ED. One of the more widely studied management methods is the application of “lean thinking” principles, so-called “lean management”. The application of this concept was tested in nine studies, analyzed by Bucci et al. Eight of the analyzed studies presented promising results of an increased number of patients served, a reduction in the duration of patient stays and an increase in patient satisfaction levels. Benefits were also observed, such as a reduction in the number of patients waiting, i.e., “left without being seen” (LWBS), and a reduction in costs. However, the authors pointed out that the application of the principles of “lean management” in a greater number of health care facilities requires in-depth research on their effectiveness first [22].

Another solution in the area of ED work organization was the use of a special ERD decision support program (Emergency Room having a Decision-Support Program—ERDS program). As a result of the implementation of this program in the group of HRHC patients, a reduction in the number of hospitalizations and interventions in the ED was achieved in favor of an increase in the number of consultations provided by doctors within primary health care. As emphasized by the researchers, the use of the ERDS program made it easier for patients to find medical help at the appropriate level of the health care system. In addition, the applied solution with a total cost of US $2.75 million resulted in US $3.41 million savings and thus achieved a rate of return on investment (ROI) of 1.24 [23].

Another researched improvement positively influencing the costs of ED functioning was the launch of a professional admission holding area for patients. Despite the initial large financial outlays due to the need to provide appropriate equipment and employ additional personnel, the solution turned out to be cost-effective in the end. The analyzed solution assumed the operation of the admission room every day for only half a day and the use of 60% of its organizational resources. It was estimated that such a solution allowed for increasing the hospital’s income thanks to the possibility of providing services to twenty additional patients, which could mean increases of as much as US $28,000 a day and US $6.09 million over 8 months. Moreover, in the discussed study, the cost of hospitalization of a patient in an ED bed was calculated as twice as high as compared to another hospital ward that conducted diagnostics and treatment, and as five times as high as in the ER. Overall, the study estimated that when patients are “held” in the ED, the hospital loses in three areas. Not only it does not receive money for an empty bed in another diagnostic department, but also incurs high costs of maintaining the bed in the ED with a minimal return of funds for the procedures performed there, and what is particularly important is that it loses potential financial resources due to the inability to use the ED services by other patients in the waiting queue [24]. The above calculations were also confirmed by other studies. They indicate that as many as eight more people can get help in the hospital, thanks to the reduction of the waiting time for one patient by 30 min.

On an annual basis, this is an increase in the hospital’s revenues at the level of US $2.7–3.9 million [25]. On the other hand, Rivers et al., on the example of the Henry Ford Hospital in Detroit showed that expenses made in the area of equipment, personnel and better organization of ED work generated annual savings of US $11.5 million in the hospital budget [26].

An important aspect influencing ED maintenance costs are imaging and laboratory tests. Based on US data for 2006–2008 from the national register of services within the ED (the National Hospital Ambulatory Medical Care Survey-NHAMCS), it was found that compared with interventions in the ED based on treatment, ordering any tests during the patient’s stay in an ED increased its length, which was particularly noticeable when it comes to visiting patients in the ED completed discharge to home [27]. In this area, it is also worth citing an interesting analysis by Bodansky et al. They showed that in as many as 40% of cases, ED doctors did not re-order tests that were initially ordered but proved impossible to be performed in the laboratory from the samples, which might suggest that routine testing is unnecessary in the first place [28]. These results confirm the validity of recommendations for prudent commissioning of laboratory tests appearing in other studies dealing with cost reduction in emergency departments.

An important analysis from the point of view of this study is the Flores-Mateo et al. study analyzing data from 48 studies relating to various solutions that improve the functioning and financing of emergency departments, which concluded that an effective solution improving the functioning of ED is the simultaneous use of patient co-payment for services received in the ED and the improvement of access to primary care. It was also established that the possible application of the other analyzed interventions should primarily take into account the impact on the health of individual patients and the proper spending of funds in the health care system, as there is no clear evidence of their effectiveness [29].

Considering the effectiveness of patient co-payment for health services in ED, it is worth citing the study by Jung et al. from South Korea, where the co-payment mechanism is present in the form of an emergency fee. The study showed that increasing the fee resulted in only a 2.4% reduction in the percentage of non-urgent ED, which resulted in a 5-min reduction in the time of patients staying in the ED [30].

Another noteworthy work is the already cited analysis of as many as 38 studies on the use of the co-payment mechanism on the part of patients and other cost optimization strategies in the ED. The authors obtained inconclusive results regarding the effect of co-payments on the reduction of patient visits to the ED, at the same time demonstrating the effectiveness of solutions based on the principles of “case management ” [31].

In turn, Morgan et al. demonstrated that in as many as nine out of the ten analyzed studies conducted in the US, a co-payment for health services obtained in the ED statistically significantly reduced the number of ED visits by as much as 35% to 50% [32].

Therefore, as mentioned before, the actual effectiveness of the use of co-payments by the patient in the ED and its impact on the improvement of the financial condition of health care requires further research to establish an unequivocal position. It is worth noting, however, that the introduction of patient co-payment in the form of the so-called “emergency fee” may be one of the few mechanisms reducing the ED burden by reducing visits by patients eligible for treatment in primary health care, the evening and night care ward, or hospital wards, which would facilitate the adequate use of financial and human resources in the health care system. The co-payment mechanism can also educate patients by increasing individual responsibility for emergency care and discouraging the use of ED services in non-life-threatening situations. 

Flores-Mateo et al. also assessed the effectiveness of an intervention by providing educational guidance to patients and supporting the development of self-care. The authors concluded that the most effective education is related to a specific disease entity or a complex multi-functional intervention, e.g., involving health education, learning to use the health care system and social counseling [31].

In a systematic review by Morgan et al., five studies looked at the effectiveness of educational interventions in the US, but only two of them showed a statistically significant impact of this education on the number of ED applications [32].

Systemic solutions aiming to reduce the number of non-urgent visits to the ED should be based mainly on coordination and integration of primary care, ambulatory specialistic cere and hospital care, improving the quality and availability of services within the health care services provided at night and during holidays, as well as properly utilized, large-scale, case management and patient navigation systems. There is insufficient evidence that the use of interventions such as pre-hospital emergency telemedicine solutions or patient co-payment will have the desired effect.

Based on the analysis carried out in this study, it can be concluded that the solution consisting in the introduction of co-payment on the part of patients who do not require hospitalization after ED provision is not a long-term solution for improving the financial situation of ED. It can only be educational for patients by increasing their level of awareness in the event of the need for an appropriate form of assistance in the event of a sudden health emergency or deterioration of health.

The limitations of this study include, first of all, the fact that it is not possible to directly translate the ED financing solutions used in the analyzed countries to the conditions in Poland. This is due to, on the one hand, differences in socio-economic conditions, for example the much larger population of Poland in comparison to Latvia. Other conditions that differ between the countries studied include, but are not limited to, underlying health profiles of different nationalities, different socioeconomic characteristics, stoicism of population and the skill level of providers. The Polish national health care system, in particular in the field of hospital care, is structured and financed differently than in the analyzed countries, for example, when compared to Germany, Poland has a much lower expenditure on health care as a proportion of the GDP.

However, the most important limitation of the study presented in this paper is, first of all, the inability to precisely determine the value of individual medical procedures performed in the ED used in the calculations. This is mainly due to the fact that while the scoring of individual outpatient specialist care services based on generally available data and regulations is relatively easy to perform, such a detailed valuation of individual services available only in the field of hospital treatment is difficult. The main reason for these difficulties is the inability to separate from a given group of hospital services only one specific procedure, which is usually part of a much larger and more comprehensive health service settled under the diagnosis-related group reimbursement methodology (in Polish: *Jednorodne Grupy Pacjentów, JGP*).

## 5. Conclusions

The introduction of a “barrier mechanism” in the form of co-payment by people reporting to ED with ailments that do not require hospitalization may be of an educational value in terms of raising patients’ awareness as to the proper choice of a place of assistance in the event of a health emergency. The key element is to relieve the ED by reducing the number of patients presenting with minor health problems and preventing overcrowding. Reduction of the ED burden in the above-mentioned categories of patients may occur, inter alia, by moving them to an ER or the evening and night care wards, which should ultimately be located in the vicinity of the ED, as well as proper coordination and logistics of patient transport by medical services through adequate and extensive use of ICT systems. A long-term solution lies in educational and information campaigns aimed at patients regarding the proper choice of the form of assistance depending on ailments, patients’ health and the risk of life-threatening conditions. Considering the possibility of taking this type of action seems to be justified both by local government authorities, which are to a large extent the founding bodies for hospitals with ED structures in Poland, and by government authorities, in particular voivodes, who exercise direct supervision over the functioning of the system on the governed territory. 

## Figures and Tables

**Table 1 ijerph-19-01507-t001:** Estimated value of services provided in the UCC in the reporting periods in 2017–2019 according to the average value of individual categories of the patient’s health condition in ED (in $) ^1^.

Patient Severity Category in the ED	Year			
	2017	2018	1st Half of 2019	2nd Half of 2019
I	$5256.33	$949.33	$818.23	$276.43
II	$501,654.48	$379,007.67	$415,703.77	$445,598.28
III	$1,463,926.90	$2,620,792.84	$2,770,816.95	$2,838,855.90
IV	$317,027.61	$427,159.02	$549,517.91	$1,208,063.28
V	$214,521.66	$245,133.68	$181,921.30	$175,812.00
VI	$67,774.86	$68,159.77	$147,071.42	$317,911.97
TOTAL	$2,570,161.84	$3,741,202.25	$4,065,849.58	$4,986,517.88

^1^ Author’s own compilation.

**Table 2 ijerph-19-01507-t002:** Comparison of the value of services provided in CED according to the average value of patient severity category in ED and the value of the contract based on daily lump sums granted to CED by NHF PVB) in the reporting periods in 2017–2019 [6] ^1^.

Year	The Value of the Contract Is Based on a Daily Lump Sum from the NHF ^2^	Estimated Value of the Services Provided	% Difference
2017	$1,832,057.94	$2,570,161.86	29%
2018	$2,128,290.43	$3,741,202.29	43%
1st half of 2019	$1,093,512.48	$4,065,849.58	73%
2nd half of 2019	$1,419,066.50	$4,986,517.86	72%
Average difference in%			54%

^1^ Author’s own compilation, ^2^ The total value of the contract in 2019 was $2,512,578.98.

**Table 3 ijerph-19-01507-t003:** Estimated value of services provided in the UCC in the reporting periods in 2017–2019 according to the average value of individual categories of the patient’s health condition in ED (in $) ^1^.

Model	Year			
	2017	2018	First Half of 2019	2nd Half of 2019
Model A	$11,105,617.54	$12,769,978.82	$9,723,527.48	$11,164,607.74
Model B	$2,828,611.45	$4,193,047.12	$4,307,309.47	$5,227,632.35
Model C	$5,951,221.49	$6,618,427.76	$5,679,779.65	$6,739,416.06
Model D	$1,887,590.10	$2,187,411.84	$1,129,565.07	$1,455,119.10
NHF PVB	$1,832,057.94	$2,128,290.43	$1,093,512.48	$1,419,066.50

^1^ Author’s own compilation.

**Table 4 ijerph-19-01507-t004:** Differences between the values obtained for the analyzed models and the values of the contracts with the NHF PVB for services provided by the CED in 2017–2019.

Model	Year				Average in%
	2017	2018	1st Half of 2019	2nd Half of 2019	
Model A	506%	500%	789%	687%	621%
Model B	54%	97%	294%	268%	178%
Model C	225%	211%	419%	375%	308%
Model D	3%	3%	3%	3%	3%

## Data Availability

Data sharing not applicable.

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
