# Peer review of "Searching for the Optimal Method of Financing Hospital Emergency Departments—Comparison of Polish and Selected European Solutions"

_ijerph, 2022, doi:10.3390/ijerph19031507_

Round 1

Reviewer 1 Report

The authors should not use acronyms in the abstract. For example, KOR is used for Clinical Emergency Department which seems to be the Polish acronym and not the English one which should be CED.

Also, POW NFZ is used, but not defined. Again, I suggest not using any acronym in the abstract and ensure all acronyms are defined when first use. NFZ is defined at the third appearance and stands for  National Health Fund which should instead be NHF.

Reviewer 2 Report

see file

Reviewer 3 Report

There is no empirical approach, or methodology of the models applied and explained in the results of the paper. In order to review the work, it is necessary to create a section on methodology and variables.

On the other hand, it is necessary to enter a description of the CGM CLININET HIS IT database.

Reviewer 4 Report

General remarks

In simple terms, the aim of the manuscript is to try to find ways to better finance hospital emergency departments in Poland, using as examples of comparison the cases of the Czech Republic, Germany and Latvia..

Specific remarks

Without further ado, my recommendations are as follows:

  • That the title of the manuscript is not so generic and that, at least, it mentions that it is a study for Poland;
  • The authors should (clearly) show that the cases of the Czech Republic, Germany and Latvia could indeed apply in the case of Poland. Otherwise, it does not seem to me that much can be learned, in terms of public health policies, by comparing realities, which are necessarily different, without any possibility of becoming minimally similar;
  • That the concluding section contains the limitations of the study;
  • That the manuscript is subject to a review that eliminates duplicate blank spaces, the existence of capital letters, which must be lowercase, etc..

Round 2

Reviewer 3 Report

No comments. 

Reviewer 4 Report

I have no more reccomendations.